# Cardiac Surgery Patients Have Reduced Vascularity and Structural Defects of the Retina Similar to Persons with Open-Angle Glaucoma

**DOI:** 10.3390/diagnostics14050515

**Published:** 2024-02-29

**Authors:** Gabija Vičaitė, Liveta Barišauskaitė, Viktorija Bakstytė, Brent Siesky, Alice Verticchio Vercellin, Ingrida Janulevičienė

**Affiliations:** 1Medical Academy, Lithuanian University of Health Sciences, Eiveniu 2, LT-50161 Kaunas, Lithuania; liveta.barisauskaite@stud.lsmu.lt (L.B.); viktorija.bakstyte@gmail.com (V.B.); ingrida.januleviciene@kaunoklinikos.lt (I.J.); 2Department of Ophthalmology, Icahn School of Medicine at Mount Sinai, New York, NY 10029, USA; brent.siesky@mssm.edu (B.S.); alice.verticchio@mssm.edu (A.V.V.)

**Keywords:** open-angle glaucoma, perfusion, optical coherence tomography, optical coherence tomography angiography

## Abstract

(1) Background: Growing evidence suggests impairment of ocular blood flow in open-angle glaucoma (OAG) pathology, but little is known about the effect of an impaired cardiovascular supply on the structural and vascular parameters of the retina. This study aims to investigate the variations of these parameters in OAG patients compared to patients undergoing cardiac surgery (CS) with cardiopulmonary bypass. (2) Methods: Prospective observational study with 82 subjects (30 controls, 33 OAG patients, and 19 CS patients) who underwent ophthalmological assessment by swept-source OCT and CDI in one randomly selected eye. (3) Results: In the CS group, OA and SPCA PSV and EDV were significantly lower, OA and SPCA RI were significantly higher compared to the OAG and healthy subjects (*p* = 0.000–0.013), and SPCA EDV correlated with linear CDR (r = −0.508, *p* = 0.027). Temporal ONH sectors of GCL++ and GCL+ layers in the CS group did not differ significantly compared to the OAG patients (*p* = 0.085 and *p* = 0.220). The CS patients had significantly thinner GCL++ and GCL+ layers in the inner sectors (*p* = 0.000–0.038) compared to healthy subjects, and these layers correlated with the CRA PSV, EDV, and RI and SPCA PSV (*p* = 0.005–0.047). (4) Conclusions: CS patients had lower vascular and structural parameters in the ONH, and macula compared to the healthy controls that were similar to persons with OAG.

## 1. Introduction

Open-angle glaucoma (OAG) represents a multifactorial progressive optic neuropathy, characterized by retinal ganglion cell (RGC) death, distinctive morphological changes to the optic nerve head (ONH), and thinning of the retinal nerve fiber layer (RNFL). Glaucomatous damage leads to distinctive visual field defects, which can progress to irreversible vision loss [1,2]. OAG is a leading cause of irreversible blindness worldwide and, despite years of research, its complex pathogenesis has not yet been fully elucidated [3,4].

Elevated intraocular pressure (IOP) is the only modifiable risk factor for OAG, and its reduction by medical and surgical means represents the mainstay of glaucoma treatment [4,5,6]. Elevated IOP can lead to compression of the optic nerve fiber bundles in the ONH, which interrupts its axoplasmic flow and causes the death of damaged RGCs [7,8]. However, elevated IOP does not explain glaucoma onset in patients with normal IOP (normal tension glaucoma), or disease progression in glaucoma patients with adequate IOP control [9,10]. Growing evidence suggests that ocular and systemic vascular risk factors play an important role in OAG development [4,10]. A large body of evidence demonstrated that a reduction in ocular blood flow may cause ischemic damage and lead to the death of RGCs and structural changes in the ONH [6,11]. Furthermore, when ocular perfusion pressure (OPP; calculated from blood pressure and IOP) decreases, impaired vascular autoregulation of the retina may lead to poor retinal microvascular perfusion and ultimately disease progression [12,13]. Instability in OPP has also been shown in persons with systemic cardiovascular diseases, including diabetes mellitus, arterial hypertension, and nocturnal dipping, thus suggesting a potential elevated co-morbidity risk for OAG [4,7,14].

Optical coherence tomography (OCT) is widely used to assess characteristic glaucomatous structural changes in the ONH and macular regions (including assessment of the thickness of the RNFL, ganglion cell layer (GCL)+ (GCL + inner plexiform layer [IPL]), and GCL++ (RNFL + GCL + IPL) layers, macular ganglion cell complex (GCC) and ONH parameters) [3,15,16]. OCT angiography (OCT-A) provides additional information about vascular parameters, including vessel density (VD) and perfusion density in the retina and choroid [7,17,18]. Reduced OCT-A-derived peri-papillary vascular parameters are strongly associated with structural and functional glaucomatous changes, and studies have shown that they may be early predictors of disease before structural changes [19,20]. Little is known about changes in ocular blood flow in patients with cardiovascular diseases as compared to glaucoma. Therefore, the aim of this study was to investigate systemic, retinal, and ONH vascular parameters assessed with multiple imaging modalities, including OCT-A and color Doppler imaging (CDI), in patients undergoing cardiac surgery (CS) with cardiopulmonary bypass compared to OAG patients and healthy controls. The relationships between ocular and systemic parameters in the three groups were also analyzed.

## 2. Materials and Methods

This prospective observational single-visit clinical study was conducted in the Department of Ophthalmology, Lithuanian University of Health Sciences Hospital, Kaunas Clinics (ClinicalTrials.gov Nr. NCT04943458). The study was approved by the Institutional Ethics Committee on 1 April 2021 (nr. 1973714). All participants signed a written informed consent form before taking part in this study.

The inclusion criteria were: age over 18 years; control group: healthy volunteers with no OAG, no acute or uncompensated chronic disease from anamnesis or at the time of the examination that could affect the results of the study; OAG group: patients with a confirmed diagnosis of open-angle glaucoma, defined as characteristic optic nerve disc (OND) changes and visual field loss consistent with glaucoma using Humphrey (24-2) SITA-FAST), and IOP controlled with treatment; CS group: patients undergoing planned cardiac surgery with a cardiopulmonary bypass as diagnosed by experienced cardiac surgeon. Exclusion criteria: subjects unable to provide written informed consent; patients allergic or sensitive to local anesthetics; patients with a history of severe ophthalmological diseases (including trauma, past ocular surgeries, age-related macular degeneration, retinitis pigmentosa, and high-grade refractive errors [astigmatism up to ±3.0 D, myopia −6.0 D or higher, and hyperopia up to +6.0 D]) that could affect the results of the study; patients with acute or uncontrolled systemic diseases; patients with diabetes mellitus. 

All patients underwent a comprehensive medical and ocular examination. Patients’ demographic profiles were recorded, measurements of height and weight were taken, and body mass index (BMI) was calculated; blood pressure (BP) and heart rate (HR) measurements were performed. One eye per patient was chosen for evaluation using randomized selection, taking into account the quality of the test results. Patients underwent a complete ophthalmic examination including best-corrected visual acuity, Goldmann applanation tonometry, and slit lamp examination. CDI (Mindray M7, Shenzhen, China) was used to obtain retrobulbar blood flow parameters of peak systolic velocity (PSV) and end-diastolic velocity (EDV) in the ophthalmic artery (OA), central retinal artery (CRA), and short posterior ciliary arteries (SPCA). The resistance index (RI) in the respective arteries was calculated using Porcelot’s formula: RI = (PSV − EDV)/PSV. A swept-source OCT (DRI-OCT Triton, Topcon, Tokyo, Japan) was used to assess ONH and macular structural and vascular parameters. The measured ONH structural parameters included rim area, disc area, linear and vertical cup-to-disc ratio (CDR), and cup volume. The thicknesses of RNFL, GCL++ (comprised of RNFL, GCL, IPL), GCL+ (including GCL and IPL), and choroidal stroma (CS) layers were measured in the peripapillary and macular retina. The Early Treatment of Diabetic Retinopathy Study (ETDRS) grid was used for the assessment of the structural parameters at a regional level in the ONH and macula. In the peripapillary area (6 × 6 mm ONH scan), the ONH parameters were assessed in four quadrants (superior (S), nasal (N), inferior (I), and temporal (T)) and distributed in twelve clock-hour independent sectors (Figure 1A,B). The macular parameters (7 × 7 mm scan) were assessed in seven sectors: center or fovea (C), inner and outer superior (IS; OS), inner and outer nasal (IN; ON), inner and outer inferior (II; OI), inner and outer temporal (IT; OT) sectors (Figure 1C). 

VD values were evaluated in the ONH and macula using 3 × 3 mm OCT-A scans (DRI-OCT Triton, Topcon, Tokyo, Japan). The OCT-A images were centered on the disc and fovea, allowing the alignment and superimposition with the infrared fundus image yielded by SS-OCT using the ImageJ software (Version 1.53k, National Institutes of Health, Bethesda, ML, USA). The VD was calculated using the ImageJ software and defined as the percentage (%) area occupied by capillaries. Four different layers of the retina and choroid were assessed, including superficial capillary plexus (SCP), deep capillary plexus (DCP), avascular retinal zone, and choriocapillaris layer. The Garway–Heath grid was used to evaluate VD in the ONH, composed of the center (C) and six sectors around the ONH: superior temporal and nasal (ST; SN), N, inferior nasal and temporal (IN; IT), and T (Figure 2A). In the macula, a grid of five sectors was applied, which included the C, S, N, I, and T quadrants (Figure 2B). Each parameter was measured three times in each grid region and the mean value was used to ensure accuracy.

Statistical analyses were performed using the Microsoft Office Excel 2021 program, IBM SPSS Statistics, version 29.0.1.0. Descriptive values were represented as the mean ± standard deviation (SD) for continuous variables and frequencies (in percent) for categorical variables. The distribution of continuous variables was assessed using the one-sample Kolmogorov–Smirnov test of normality. Factors with normal distribution by the KS test were compared using the independent-samples one-way ANOVA test. The Pearson two-tailed correlation test was used to determine the strength of the correlation between those variables. Factors with abnormal distribution were compared using the independent-samples Kruskal–Wallis one-way ANOVA test (significance values adjusted by the Bonferroni correction for multiple tests). The strength of the correlation between these variables was assessed using the Spearman two-tailed correlation test. Differences in categorical variables were detected using the Pearson Chi-Square test. The differences were considered statistically significant with a *p*-value of <0.05. The correlation was considered weak when r was ≤0.3, medium when 0.3 < r ≤ 0.7, and strong when 0.70 < r ≤ 1.

## 3. Results

### 3.1. Demographic Characteristics and Systemic Parameters

A total of 82 subjects (30 healthy controls, 33 OAG patients, 19 CS patients) were recruited and 82 randomly selected eyes were examined in the study. Table 1 shows the demographic characteristics and systemic parameters of the study participants. The patients in the CS group were significantly older compared to the OAG and control groups (*p* < 0.001). The average BMI of the study sample was 27.32 ± 4.77 kg/m^2^. A total of 63.4% (52 out of 82) of study participants were above a normal BMI (33 (40.2%) overweight (BMI > 25 kg/m^2^), and 19 (23.2%) were obese (BMI > 30 kg/m^2^)). The average BP (systolic BP/diastolic BP) was 136.94 ± 18.70/80.35 ± 11.86 mmHg and the mean HR was 69.55 ± 12.09 bpm. Mean diastolic BP and HR in the CS group were higher compared to the other groups, with no statistically significant difference.

### 3.2. Color Doppler Imaging

No significant differences were found between the retrobulbar blood flow parameters in the OAG patients compared to the healthy controls (*p* > 0.05). Several significant differences were established between the OAG and CS groups, as presented in Table 2. Mean PSV and EDV were significantly lower and mean RI was significantly higher in the OA and SPCA of the CS patients group compared to the OAG and healthy subjects (*p* = 0.000–0.013). The CRA EDV was significantly lower and RI was significantly higher in the CS patients group compared to the OAG and healthy subjects (*p* < 0.001). CRA PSV was lower but did not significantly differ when compared to glaucoma patients and healthy controls (*p* > 0.05).

### 3.3. OCT and OCT-A

#### 3.3.1. Optic Nerve

All structural ONH parameters significantly differ in the OAG group compared to the other groups (Table 3). In CS patients, the neuroretinal rim area was significantly smaller than in healthy controls (*p* = 0.015). The linear CDR in the CS group was higher than the control (0.53 ± 0.21 and 0.44 ± 0.26, respectively), and it was negatively associated with the SPCA EDV (r = −0.508, *p* = 0.027).

In OAG patients, RNFL (72.71 ± 22.00), GCL++ (107.27 ± 29.96), and GCL+ (34.71 ± 10.32) thicknesses were significantly lower than in healthy controls (104.37 ± 7.68, *p* = 0.000; 146.86 ± 10.80, *p* = 0.000; 42.33 ± 4.89, *p* = 0.004, respectively), especially in the N and T quadrants. RNFL, GCL++, and GCL+ layer thicknesses were positively and significantly associated with OA EDV (r = 0.366, r = 0.449, and r = 0.447, respectively). In the CS group, RNFL, GCL++, and GCL+ thicknesses were lower than in the control group, especially in the S, I, and T quadrants, though no statistical significance was achieved (*p* = 0.112–1.000). The GCL++ and GCL+ layers’ thicknesses in the T sectors did not significantly differ between the CS and OAG groups (*p* = 0.085 and *p* = 0.220, respectively). The CS thickness did not significantly differ between the three groups (*p* > 0.05).

The OCT-A VD in the superficial and deep capillary plexus and the outer avascular zone were significantly lower in the OAG and CS group compared to healthy controls (*p* = 0.000–0.044). The CS patients had significantly lower VD in all sectors compared to the healthy controls, except for the IT sector of the superficial plexus (*p* = 0.067) and the T sector of the outer avascular zone (*p* = 0.172) (Figure 3). The VDs in the central region of the ONH choriocapillaris layer were significantly higher in the OAG (*p* = 0.001) and CS (*p* = 0.005) groups compared to the control group. In the OAG group, several moderate positive correlations were observed between RNFL, GCL++ (all quadrants), and GCL+ (I, T quadrants) thicknesses and VD in the superficial, deep plexus, and avascular layer (r = 0.356–0.689, *p* = <0.001–0.042). Both in the OAG and CS groups, moderate and strong statistically significant positive correlations were found in the T quadrant of the ONH between the deep plexus and avascular zone (OAG: r = 0.763; CS: r = 0.811), and between the avascular zone and the choriocapillaris layer (OAG: r = 0.665; CS: r = 0.689), all *p* = <0.001. Figure 3 displays the ONH VD in the three groups.

#### 3.3.2. Macula

In our study, whole macular thickness (excluding the foveal area) was significantly thinner in the OAG and CS groups compared to the healthy control (*p* = 0.000–0.018). OAG patients had significantly thinner RNFL (except in the IS, IN, and IT sectors), GCL++, and GCL+ layers (*p* = 0.000–0.016) compared to healthy subjects (Figure 4). The CS patients had significantly thinner GCL++ and GCL+ layers in the IS, IN, II, and IT sectors around the foveal area (*p* = 0.000–0.038) compared to healthy subjects, with no significant differences in the RNFL thickness (*p* > 0.05). In the CS group, statistically significant correlations were found between the thickness of the full retina, GCL++, and GCL+ layers in the inner sectors with the CRA PSV, EDV, and RI and SPCA PSV (*p* = 0.005–0.047). Figure 4 shows the thicknesses of the retinal layers in the macular region in the three groups. OAG eyes had a thicker CS in all sectors of the macula compared to the CS and control groups, with a statistically significant difference only in the OT sector (244.58 ± 98.53 µm, 186.47 ± 57.49 µm, and 230.50 ± 68.09 µm, respectively, *p* = 0.016). In the CS group, the choroid was thicker around the fovea and in the IT area, and thinner in the other sectors when compared to the control group.

In our study, macular VD did not differ significantly between groups (*p* > 0.05). In the CS patients, the deep capillary plexus foveal VD was significantly higher (32.92 ± 8.30%) compared to the OAG and control groups (25.85 ± 9.40%, *p* = 0.007, and 22.60 ± 6.42%, *p* = 0.000, respectively). In the CS patients, the deep capillary plexus foveal VD was significantly negatively associated with the OA EDV (r = −0.596, *p* = 0.007).

## 4. Discussion

In this study, we investigated systemic and ocular vascular parameters assessed via multi-modal imaging modalities including OCT, OCT-A, and CDI in OAG patients compared to CS patients and healthy controls. Several studies previously showed reduced PSV and EDV and increased RI in the retrobulbar vessels of OAG patients compared to healthy controls [21,22,23]. CDI has been suggested as a valuable tool to assess ocular blood flow and assist in diagnosis and progression monitoring [22,23]. In our study, the retrobulbar blood flow parameters did not significantly differ between OAG patients and healthy controls. However, we found few statistically significant differences between the CS group compared to OAG and healthy eyes, thus highlighting the impairment of the retrobulbar circulation in these patients (Table 2). Negative associations between linear CDR in the ONH and SPCA EDV of CS patients suggest that impaired blood flow in the SPCA may contribute to ONH structural changes. Similar results were described in an 18-month observational study conducted by Tobe et al., who found thinning of the ONH rim and CDR increase to be associated with higher RI and lower PSV in the SPCA of OAG patients [24]. Additionally, the Thessaloniki Eye Study found an association between aggressive antihypertensive therapy and higher CDR, which could be important in the interpretation of the results in CS patients [25].

ONH structural changes are characteristic of OAG patients, even before visual field damage can be detected [26,27]. Verticchio Vercellin et al. showed several characteristic structural ONH changes in patients with pre-perimetric OAG compared to healthy subjects, including significantly smaller rim area (0.87 ± 0.28 mm^2^ and 1.41 ± 0.26 mm^2^, respectively, *p* < 0.001), larger vertical CDR (0.72 ± 0.15 and 0.47 ± 0.16, respectively, *p* = 0.043), and cup volume (0.36 ± 0.30 mm^3^ and 0.11 ± 0.12 mm^3^, respectively, *p* = 0.007) [28]. Similarly, in our study, all structural ONH parameters differed significantly in the OAG group compared to the other groups (Table 3). CS had a smaller neuroretinal rim area and thinner GCL++ and GCL+ layers in the T quadrant of the ONH compared to healthy control, indicative of early structural changes in the optic nerve region. However, in our study, only the OAG patients presented the characteristic ONH structural glaucomatous damage, including a significant reduction in the RNFL, GCL+, and GCL++ layers. We found positive associations between the RNFL, GCL+, and GCL++ layers’ thicknesses and OA EDV, suggesting that impaired retrobulbar circulation in OAG patients can lead to structural damage and further nervous tissue loss. Similarly, Tobe et al. described structural retinal and ONH glaucomatous progression with retrobulbar blood flow reduction over time [24].

Similar to several previous studies [29,30], we found a reduction in the OCT-A-derived retinal and ONH blood flow parameters in OAG patients compared to the control. Moreover, CS patients presented VD reduction in the ONH similar to OAG patients, in both severity and localization. These results suggest that patients with significant systemic cardiovascular disease may present in the peripapillary retinal microvasculature findings that are similar to those of OAG patients (Figure 3). Both the OAG and CS patients presented significantly higher VD in the central region of the ONH choriocapillaris layer. These findings may be related to capillary dilation in response to BP or OPP fluctuations, since the choroidal circulation does not have vascular autoregulation but has a neurogenic control [22]. In agreement with our study, Khayrallah et al. and Yospon et al. found positive correlations between ONH VD and RNFL thickness in OAG patients, thus suggesting that impaired retinal and ONH microcirculation is associated with nervous tissue thinning [31,32]. We also found significant positive correlations between the deep plexus and avascular zone (CS: r = 0.811, *p* < 0.001; OAG: r = 0.763, *p* < 0.001), and the avascular zone and choriocapillaris layer (CS: r = 0.689, *p* = 0.001; OAG: r = 0.665, *p* < 0.001) in the T quadrant of the ONH in both the OAG and CS groups. These results suggest an interplay between the choroidal and retinal circulations, which are both important for optic nerve health and disease development [33].

Multiple studies showed the potential role of macular structural parameters as early predictors of glaucomatous damage; however, nearly 50% of macular retinal ganglion cells may be lost before the visual field loss becomes detectable [28,34]. Verticchio Vercellin et al. found that macular thicknesses both at the global and regional levels were significantly lower in patients with pre-perimetric OAG when compared to healthy subjects [28]. In our study, we found a similar macular thickness decrease (excluding the foveal area) in both the OAG and CS groups compared to healthy subjects. Importantly, the thinning of macular layers occurred with different regional distributions in these two groups (Figure 4). The CS group presented statistically significant correlations between the thickness of GCL++ and GCL+ and CRA and SPCA blood flow parameters. These results indicate that structural damage at the level of the macula is associated with impaired retrobulbar circulation. These correlations were not found in the OAG patients who presented different anatomical location of the macular thinning, which indicates that the pathogenetic mechanisms behind structural damage and hemodynamic impairment in the two groups may differ. Previous studies have shown that OCT-A-derived vascular macular parameters are reduced in OAG patients and have good diagnostic accuracy for glaucoma. Decreases in parafoveal VD in OAG and NTG patients have been recently reported [35]. However, in our experience, the macular VD was distinctly less differentiated between groups compared to ONH VD. Higher VD in the foveal deep capillary plexus of CS patients and its association with OA EDV may suggest that the relationship between hemodynamic and structural parameters is different among the three groups.

Our study had several limitations to acknowledge. First, we included a relatively small sample size of cardiac patients who were significantly older than the OAG and healthy groups. The statistical analysis was not adjusted to control for age, which could potentially influence the results of this study. Also, the CS patients included in our study were under various hypertensive therapies; the effect of antihypertensive therapy on BP and ocular blood flow biomarkers was not investigated. Finally, all our study participants were of European descent (Eastern European), thus, our results may not be generalized to persons of African and Asian descent. Future research should confirm findings in age-balanced groups and in those harmonized for therapies where possible.

In conclusion, our study shows that CS patients have reduced ONH VD similar to those with OAG when compared to healthy controls. However, in the OAG and CS patients, the topographic location of the structural damage at the level of ONH and macula was different. Also, the relationship between structural and hemodynamic parameters differed in the two groups, suggesting potentially different impacts from reductions in VD, possibly due to faulty autoregulation of blood flow in the OAG group. Together, this data demonstrates that significant systemic cardiovascular disease may elevate the risk for OAG, although intact vascular regulation in otherwise healthy individuals may provide a protective effect. Alternatively, the loss of VD in cardiovascular and glaucomatous disease may be due to distinct pathophysiologic mechanisms. Further investigation with a larger sample size and prospective longitudinal monitoring is warranted. Our recommendation regarding the results of this study is for clinicians to keep in mind the importance of non-IOP-related risk factors when managing glaucoma, as they might influence the progression of the disease.

## Figures and Tables

**Figure 1 diagnostics-14-00515-f001:**
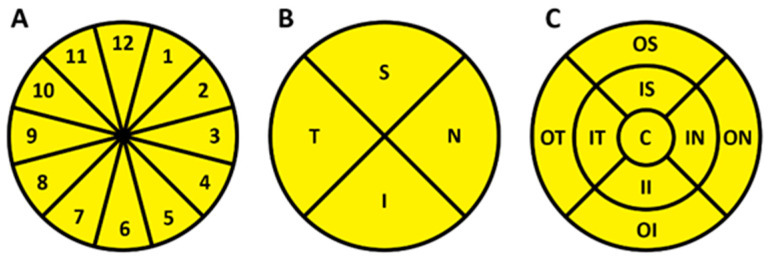
The Early Treatment of Diabetic Retinopathy Study (EDTRS) grid for optic nerve head (**A**,**B**) and macular (**C**) evaluation. (**A**)—twelve clock-hour ONH sector distribution; (**B**)—ONH quadrants: superior (S), nasal (N), inferior (I), temporal (T); (**C**)—macular sectors: center or fovea (C), inner and outer superior (IS; OS), inner and outer nasal (IN; ON), inner and outer inferior (II; OI), inner and outer temporal (IT; OT).

**Figure 2 diagnostics-14-00515-f002:**
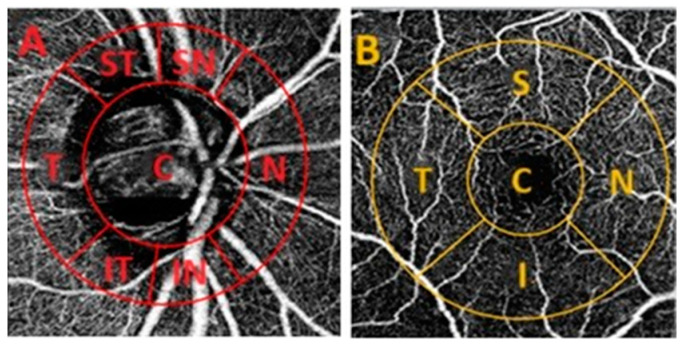
Optical coherence tomography angiography of optic nerve head (**A**, red) and macula (**B**, yellow). (**A**)—the ONH Garway–Heath grid, sectors: center (C), superior temporal (ST), superior nasal (SN), nasal (N), inferior nasal (IN), inferior temporal (IT), temporal (T); (**B**)—macular sectors: center or fovea (C), superior (S), nasal (N), inferior (I), temporal (T).

**Figure 3 diagnostics-14-00515-f003:**
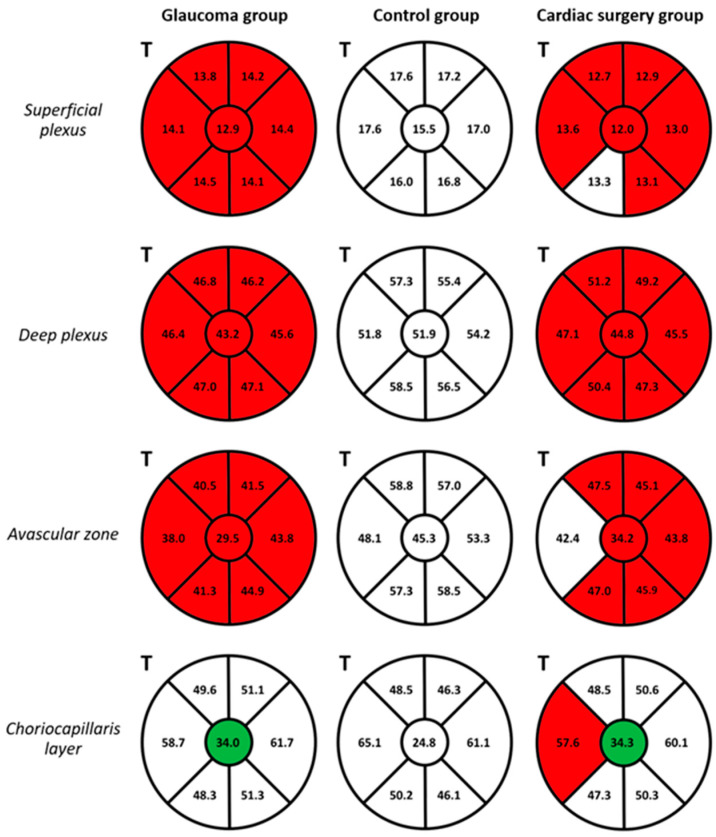
Vessel density (VD) in the optic nerve head assessed by optical coherence tomography angiography. The VD mean values are shown for each sector (temporal side (T) on the left) for the open-angle glaucoma (OAG, left), control (center), and cardiac surgery (CS, right) groups. The color of the sectors indicates the differences between the group (OAG or CS) compared to the control group. Green color: VD in the corresponding sector is significantly higher than the control group; red color: VD in the corresponding sector is significantly lower compared to the control group; white color: VD in the corresponding sector does not differ significantly compared to the control group. The differences were considered statistically significant with a *p*-value of <0.05.

**Figure 4 diagnostics-14-00515-f004:**
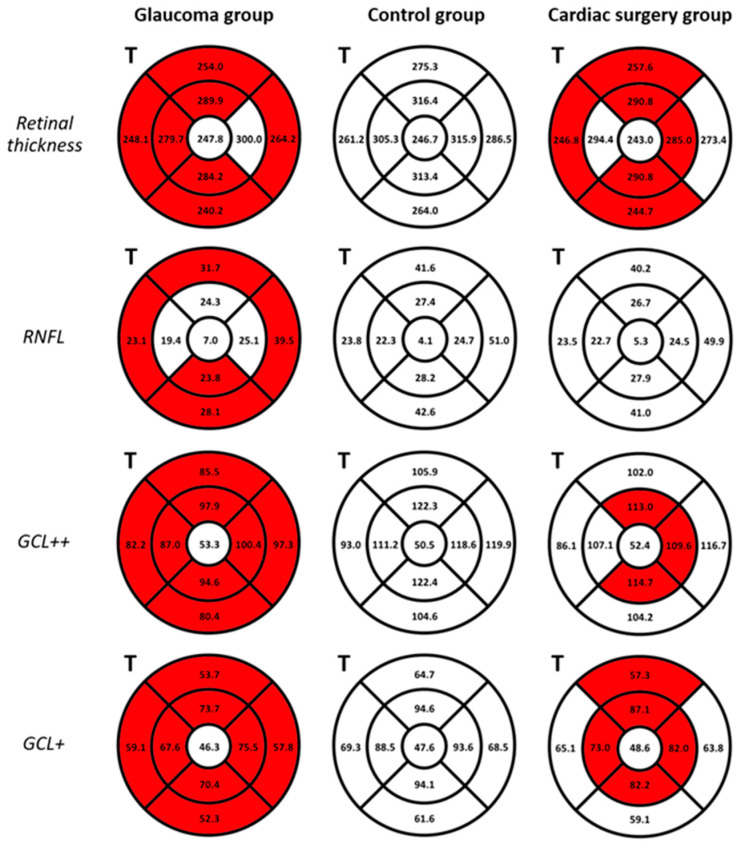
Thicknesses of the retinal layers in the macular region assessed by optical coherence tomography. The thickness mean values of the retinal layers are shown for each sector (temporal side (T) on the left) for open-angle glaucoma (OAG, left), control (center), and cardiac surgery (CS, right) groups. The color of the sectors indicates the difference between the group (OAG or CS) compared to the control group. Red color: thickness of the retinal layer in the corresponding sector is significantly lower compared to the control group; white color: thickness of the retinal layer in the corresponding sector does not differ significantly compared to the control group. The differences were considered statistically significant with a *p*-value of <0.05. GCL: ganglion cell layer; GCL+: ganglion cell layer+ (GCL + inner plexiform layer); GCL++: ganglion cell layer++ (RNFL + GCL + inner plexiform layer); RNFL: retinal nerve fiber layer.

**Table 1 diagnostics-14-00515-t001:** Study participants’ demographic characteristics and systemic parameters. Values represent the mean ± standard deviation (SD). The differences were considered statistically significant with a *p*-value of <0.05. dBP: diastolic blood pressure; HR: heart rate; OAG: primary open-angle glaucoma; sBP: systolic blood pressure.

Parameter		Mean ± SD				*p*-Value		
		All Study Subjects (*N* = 82)	OAG Group (*N* = 33)	Control Group *(N =* 30)	CS Group (*N* = 19)	OAG vs. Control	OAG vs. CS Group	CS Group vs. Control
**Sex**	Male	37 (45.1%)	14 (42.4%)	11 (36.7%)	12 (63.2%)	0.641	0.150	0.070
	Female	45 (54.9%)	19 (57.6%)	19 (63.3%)	7 (36.8%)			
Age (years)		62.46 ± 7.13	61.24 ± 4.67	59.03 ± 5.14	70.00 ± 8.08	0.134	<0.001	<0.001
Height (cm)		169.51 ± 8.99	170.03 ± 9.84	169.80 ± 8.69	168.16 ± 8.18	0.920	0.475	0.538
Weight (kg)		78.85 ± 15.51	79.55 ± 15.91	80.13 ± 18.52	75.63 ± 8.12	1.000	1.000	1.000
BMI (kg/m^2^)		27.32 ± 4.77	27.47 ± 4.95	27.66 ± 5.40	26.51 ± 3.31	1.000	1.000	1.000
sBP (mmHg)		136.94 ± 18.70	134.18 ± 18.26	138.17 ± 19.52	139.79 ± 18.49	0.403	0.303	0.769
dBP (mmHg)		80.35 ± 11.86	80.48 ± 14.08	77.63 ± 9.03	84.42 ± 10.92	0.338	0.247	0.052
HR (bpm)		69.55 ± 12.09	68.42 ± 10.62	67.37 ± 12.72	74.95 ± 12.49	1.000	0.149	0.058

**Table 2 diagnostics-14-00515-t002:** Retrobulbar blood flow parameters assessed by color Doppler imaging. Values represent the mean ± standard deviation (SD). The differences were considered statistically significant with a *p*-value of <0.05. EDV: end-diastolic velocity; PSV: peak systolic velocity; OAG: primary open-angle glaucoma; RI: resistance index.

Artery	Parameter	Mean ± SD			*p*-Value		
		OAG Group (*N* = 33)	Control Group (*N* = 30)	CS Group (*N* = 19)	OAG vs. Control	OAG vs. CS Group	CS Group vs. Control
Ophthalmic artery	PSV (cm/s)	50.02 ± 9.89	50.87 ± 12.52	32.72 ± 8.99	0.756	<0.001	<0.001
	EDV (cm/s)	14.15 ± 3.63	15.49 ± 6.09	7.51 ± 3.22	0.256	<0.001	<0.001
	RI	0.71 ± 0.06	0.70 ± 0.08	0.77 ± 0.08	0.317	0.013	0.001
Central retinal artery	PSV (cm/s)	16.09 ± 6.55	15.83 ± 5.24	14.57 ± 6.66	1.000	0.479	0.300
	EDV (cm/s)	7.24 ± 2.34	6.97 ± 2.43	4.33 ± 1.64	1.000	0.000	0.000
	RI	0.53 ± 0.15	0.55 ± 0.09	0.69 ± 0.08	0.376	<0.001	<0.001
Short posterior ciliary arteries	PSV (cm/s)	21.50 ± 5.91	22.44 ± 3.52	15.52 ± 5.00	0.686	0.001	0.000
	EDV (cm/s)	9.80 ± 2.85	11.10 ± 2.84	4.32 ± 1.44	0.411	0.000	0.000
	RI	0.54 ± 0.11	0.51 ± 0.10	0.71 ± 0.08	0.247	<0.001	<0.001

**Table 3 diagnostics-14-00515-t003:** Optic disc structural parameters assessed by optical coherence tomography. Values represent the mean ± standard deviation (SD). The differences were considered statistically significant with a *p*-value of <0.05. CDR: cup-to-disc ratio.

Optic Disc Parameters	Mean ± SD			*p*-Value		
	OAG Group (*N* = 33)	Control Group (*N* = 30)	CS Group (*N* = 19)	OAG vs. Control	OAG vs. CS Group	CS Group vs. Control
Rim area	0.60 ± 0.44	1.53 ± 0.40	1.23 ± 0.35	<0.001	<0.001	0.015
Disc area	2.13 ± 01.22	2.04 ± 0.38	1.88 ± 0.36	0.000	0.000	0.284
Linear CDR	0.81 ± 0.18	0.44 ± 0.26	0.53 ± 0.21	0.000	0.000	0.865
Vertical CDR	0.83 ± 0.18	0.43 ± 0.22	0.56 ± 0.18	<0.001	<0.001	0.031
Cup volume	0.43 ± 0.34	0.09 ± 0.12	0.13 ± 0.14	0.000	0.001	0.760

## Data Availability

The data supporting the findings of this study are available from the corresponding author, G.V., upon reasonable request.

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
