# Peer review of "Cardiac Surgery Patients Have Reduced Vascularity and Structural Defects of the Retina Similar to Persons with Open-Angle Glaucoma"

_diagnostics, 2024, doi:10.3390/diagnostics14050515_

Round 1
Reviewer 1 Report
Comments and Suggestions for Authors
Interesting study, carefully done. A question: What are the consequences of your results for the clinician? Can you give some recommendations? Please mention this in the discussion.
Comments on the Quality of English Languageonly minor corrections
Reviewer 2 Report
Comments and Suggestions for Authors
General comments:
· Please revise all the acronyms used in the manuscript. The authors should describe the acronym for the first time and then use it throughout the manuscript. For example, the acronym for CDI (line 90), OCT (line 94), and ONH (line 95) were previously described in the introduction section. Please check this for every acronym.
Introduction:
· Lines 28-30: Please consider rewriting this sentence. It reads a bit odd since it says “characterized” and “characteristic”.
· Lines 61-64: Please consider rewriting this sentence to: “Therefore, the aim of this study was to investigate systemic, retinal, and ONH vascular parameters assessed with multiple imaging modalities, including OCTA and color Doppler imaging (CDI), in patients undergoing cardiac surgery with cardiopulmonary bypass (CS) compared to OAG patients and healthy controls”.
· Line 64: Why did the authors abbreviate cardiopulmonary bypass to CS?
Materials and methods:
· Lines 74-75: Please clarify what do you mean by “no acute or uncompensated chronic disease that could affect the results of the study”. What if a patient had a history of acute myocardial infarction, but now he/she is stable? The authors must clarify exactly which patients were eligible for the control group.
· Lines 79-83: Same comment applies for the general exclusion criteria. What do you mean by “severe ophthalmological diseases that could affect the results of the study”.
· Lines 87-88: Which randomization method did the authors used?
Results:
· Overall results: The authors should control for age the analyses presented in section 3.2 (table 2), section 3.3.1 (table 3 and figure 3), and section 3.3.2 (figure 4).
· Figures 3 and 4: I would rather present this information in tables, but I will leave this up to the authors.
Discussion:
· Limitations: The authors did not control the statistical analysis for age, which may also affected the parameters analyzed.
Comments on the Quality of English LanguageMinor English review necessary.
